# Thermal limits of bumblebees and honeybees are modulated by different functional traits: predictions of a mechanistic model

**Sarah A. MacQueen**[1,2]*, **Dara A. Stanley**[1,2], **Jon M. Yearsley**[2,3]

**1** School of Agriculture and Food Science, University College Dublin, Dublin, Ireland, **2** Earth Institute, University College Dublin, Dublin, Ireland, **3** School of Biology and Environmental Science, University College Dublin, Dublin, Ireland

* macquees@gmail.com

**Data availability statement:** All code and data files are accessible on Open Science

## Abstract

Local weather conditions are expected to have species specific effects on the activity of insect pollinators. However, the relationship between changing weather patterns and pollinator activity has not been well studied. We develop a thermodynamic model for insect thorax temperature that provides a mechanistic link between local weather conditions and functional traits (e.g. body mass, flight speed) and flight activity. We show that behavioural warming and cooling adaptations are essential for temperate bumblebees and the western honeybee, and that the maximum air temperature for sustained flight depends primarily on flight speed for honeybees, whereas for bumblebees it depends upon both flight speed and thorax mass. Our results suggest that the activity of these two pollinator groups will respond differently to climate change, and that different bee groups may provide a compensatory role for each other in different weather conditions. Thus, both are important for sustained crop pollination under future change.

## Introduction

As providers of pollination services, bees are critical in ensuring ecosystem functioning and food security [1,2]. There is a huge diversity in bee species, and differences in functional traits mean that they are not always interchangeable as pollinators [3,4]. For example, although honeybees have larger colony sizes providing large numbers of workers, bumblebees are superior pollinators for certain crops or wild plants due to differences in behaviour (e.g ability to buzz pollinate) or morphology (e.g. tongue length) which allow pollination of separate suites of plants. Honeybees and bumblebees are two key bee pollinator groups in temperate regions, providing a large proportion of crop pollination services [5]. Both groups have known differential responses to current weather conditions [6,7], suggesting that their activity will likely respond differently to future changes in weather patterns.

The large-scale effects of climate change on bees, such as phenological mismatch between bees and plants [8–10] and changes in range or distribution [11,12], have been reasonably well

Framework. https://osf.io/sajkt/ (Doi: 10.17605/OSF.IO/SAJKT).

**Funding:** DAS has been supported by Science Foundation Ireland (Grant number 17/CDA/4689 to DAS). https://www.sfi.ie/. The funder played no role in the study design, data collection and analysis, decision to publish, or preparation of the manuscript.

**Competing interests:** The authors have declared that no competing interests exist.

studied. Less well studied are the fine-scale effects (i.e. hourly or daily) of climate change on bee activity and the knock-on implications for the delivery of pollination services. Measuring these small-scale changes in the field is difficult. To address this we develop a mechanistic mathematical model of a bee's thorax temperature. Thorax temperature is a key determinant of a bee's activity, and must be within a specific range for flight activity to be possible (approximately 35-52 °C for honeybees and 30-45 °C for bumblebees)[13,14]. To model the thorax temperature, we incorporate environmental, physiological, and behavioural mechanisms of heat loss and gain into the heat equation [15], including the effects of functional traits on these mechanisms. We then solve numerically for the equilibrium thorax temperature.

Previous models of bee thermoregulation have focused on heat budgets and often include mechanisms that are empirically rather than mechanistically modelled [16,17]. Our approach extends models of thorax temperatures developed for other insects [18,19] because environmental mechanisms involved in heating and cooling remain largely the same across many different insect species. However, bees can produce large amounts of metabolic heat [20,21] and also rely on behavioural cooling mechanisms, both of which require additional model terms. For example, some bees shed heat via the abdomen or head [22–25], while others use evaporative cooling from the mouth [17,26,27]. We model the processes involved in all mechanisms of heat gain and loss specific to temperate bumblebees and the western honeybee. Since our model incorporates all known thermodynamic mechanisms we expect the predictive power of our model to extend beyond the range of the data used to parameterize it, allowing a more complete understanding of how bumblebees and honeybees respond to weather conditions (specifically air temperature, solar radiation, relative humidity, and ground temperature). We therefore use the model to suggest potential effects of future climate change on bumblebee and honeybee activity, the role played by the bee's functional traits in constraining activity, and consider the implications for pollination services.

## Methods

We have developed a model for the thorax temperature of flying insects resulting from environmental, physiological, and behavioural mechanisms (Fig 1) and as a function of the weather variables air temperature, solar radiation, relative humidity, ground temperature, ground reflectance, and the latent heat of vaporisation of water. We have parameterized the model for bumblebees (*Bombus* spp.) from temperate regions and the western honeybee (*Apis mellifera*). Although it would be ideal to obtain all parameter values from the same species, and indeed for multiple species, this is not possible for bumblebees with the data available in the literature. Our parameters are therefore drawn from a mixture of species (*B. vosnesenskii*, *B. bimaculatus*, *B. fervidus*, *B. vagans*, *B. terrestris*, *B. pascuorum*, *B. lapidarius*, *B. derhamellus*, *B. impatiens*, and *B. terricola*). A detailed listing is provided in the supporting information. In bees, the thorax temperature is the determinant of flight ability, so we only directly model the thorax temperature. The thorax loses heat to the head, which in turn loses heat to the atmosphere, so we include heat loss from the head to the atmosphere in the model. Heat loss from the thorax to the abdomen is negligible, except when a specific behavioural cooling method is in use, so we only include it in the model in that specific case. With this model, we seek to make inferences at the scale of individuals. To do so, we model the rate of change, $dT_{th}/dt$, of thorax temperature ($T_{th}$) with respect to time ($t$). We then numerically solve for the equilibrium thorax temperature of a bee that is either flying (higher metabolic rate) or resting (lower metabolic rate). For a bee with a thorax mass $m_{th}$ and using the specific heat of insect tissue, $c$,

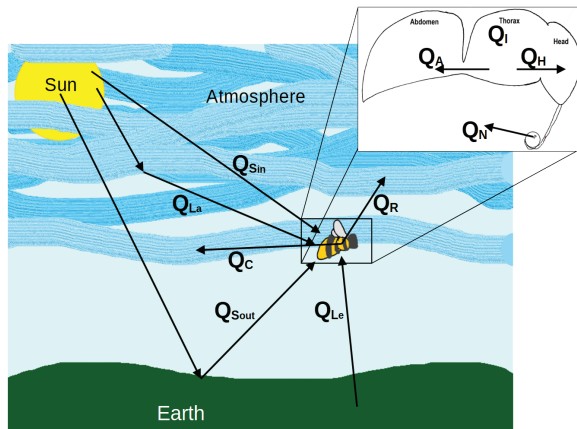

**Fig 1. Heat gain and loss mechanisms of a flying bee.** Incoming and outgoing short-wave radiation into the bee ($Q_{S_{in}}$ and $Q_{S_{out}}$); long-wave radiation from the Earth and the atmosphere into the bee ($Q_{L_e}$ and $Q_{L_a}$); thermal radiation out of the bee ($Q_R$); convective cooling ($Q_C$); movement of heat from the thorax to the head ($Q_H$); metabolic heat production within the bee ($Q_I$); cooling due to the interrupted countercurrent heat exchange between the thorax and abdomen ($Q_A$); and evaporative cooling by tongue lashing with nectar ($Q_N$). Direction of arrow indicates whether the bee is generally warmed or cooled via the indicated mechanism.

the model is

$$m_{th}c\frac{dT_{th}}{dt} = Q_S + Q_L - Q_R - Q_C + Q_I + Q_H - Q_A - Q_N. \tag{1}$$

Environmental heat flux, consisting of short-wave and long-wave solar radiation ($Q_S$ and $Q_L$, respectively), convection ($Q_C$), and thermal radiation ($Q_R$), is common to many insect thermoregulation models, and we treat it in a standard way. The physiological heat flux mechanisms of metabolic heat generation ($Q_I$) and heat transfer from the thorax to the head ($Q_H$) occur in bees [24,26], and have not previously been included in thermoregulation models for other insects. Behaviourally controlled heat flux processes seen in bees are heat transfer from the thorax to the abdomen ($Q_A$) and evaporative cooling via nectar held on the tongue ($Q_N$). To the best of our knowledge, mechanistically modelling these processes is unique to our model. Note that $Q_H$, as will be detailed later, is equivalent to the heat lost from the head to the environment and is overall a negative value. Thus we add it here instead of subtracting it as we do the other heat loss terms. Throughout the model, when there is no evidence either for or against a particular mechanism or parameter value for bumblebees, we use the same value as for honeybees, and vice versa. We provide select parameter values for bumblebees in Table 1 and for honeybees in Table 2, with the full table of parameters in Supporting information.

### Environmental heat gain mechanisms

Environmental heat gain depends variously on the solar radiation, the air temperature, and the ground temperature. Heat gain due to short-wave radiation directly from the Sun is given by $Q_{S_{in}} = \alpha_p \epsilon_a a_{th} P$, and heat gain due to short-wave radiation reflected from the Earth is given by $Q_{S_{out}} = \alpha_{np} \epsilon_a a_{th} f P$, where $\alpha_p$ is the fraction of the surface exposed to point sources of radiation (i.e. the Sun) and $\alpha_{np}$ is the fraction of the surface exposed to non-point sources of radiation (i.e. reflected sunlight from the Earth, and long-wave radiation from the atmosphere and Earth), $\epsilon_a$ is the absorptivity of the bee, $a_{th}$ is the surface area of the bee's thorax, $P$ is the

**Table 1. Key parameter values for an average temperate bumblebee (*Bombus* spp.)**

| Description | Symbol | Default value | Range | Source |
|---|---|---|---|---|
| resting metabolic rate | $i_0$ | $1.350 \times 10^{-3}$ J/s | [0.0001, 0.0082] J/s | [36] |
| active metabolic rate - literature | $i_0$ | $6.230 \times 10^{-2}$ J/s | [0.0584, 0.0662] J/s | [36] |
| active metabolic rate - fitted | $i_0$ | $6.335 \times 10^{-2}$ J/s | n/a | |
| activation energy - literature | $E$ | $0.63 \cdot 1.602 \times 10^{-19}$ J | $[0.6, 0.7] \cdot 1.602 \times 10^{-19}$ J | [31] |
| activation energy - fitted | $E$ | $0.02 \cdot 1.602 \times 10^{-19}$ J | n/a | |
| average thorax mass | $m_{th}$ | 0.057 g | [0.014, 0.132] g | [37] |
| flight speed | $v$ | 4.1 m/s | [1, 5.5] m/s | [38] |
| minimum thorax temp for flight | $T_{min}$ | 30 °C | | [13] |
| critical thermal maximum (thorax) | $CT_{max}$ | 44 °C | n/a | [22, 39] |
| thorax temperature at which cooling via the abdomen begins | $T_c$ | 42 °C | [40, 44] °C | [22] |

Note that the parameter values for bumblebees come from a mixture of different species. These species are included in a full list of parameters and values in the Supporting Information. The ranges shown for each parameter are used in the sensitivity analysis. When a range was not available from the source paper, we used a default range of ±10%, commonly used in sensitivity analyses.

**Table 2. Key parameter values for an average western honeybee (*Apis mellifera*).**

| Description | Symbol | Default value | Range | Source |
|---|---|---|---|---|
| resting metabolic rate | $i_0$ | $4.52 \times 10^{-4}$ J/s | ±10% | [40, 41] |
| active metabolic rate - literature | $i_0$ | $3.20 \times 10^{-2}$ J/s | $[4.52 \times 10^{-4}, 4.8 \times 10^{-2}]$ J/s | [40, 41] |
| active metabolic rate - fitted | $i_0$ | $3.35 \times 10^{-2}$ J/s | n/a | |
| activation energy - literature | $E$ | $0.63 \times 1.602 \cdot 10^{-19}$ J | $[0.6, 0.7] \cdot 1.602 \times 10^{-19}$ J | [31] |
| activation energy - fitted | $E$ | $0.008 \cdot 1.602 \times 10^{-19}$ J | n/a | |
| average thorax mass | $m_{th}$ | 0.041 g | SD = 0.003 g | [17] |
| flight speed | $v$ | 5.6 m/s | SD = 1.0 m/s | [42] |
| minimum thorax temp for flight | $T_{min}$ | 35 °C | n/a | [14] |
| critical thermal maximum (thorax) | $CT_{max}$ | 51 °C | n/a | [43–45] |
| head temperature at which evaporative cooling begins[a] | $T_c$ | 45 °C | [44, 46] °C | [14, 26] |

The ranges shown for each parameter are used in the sensitivity analysis. When a range was not available from the source paper, we used a default range of ±10%, commonly used in sensitivity analyses. A full list of values is available in the Supporting information.
[a]Corresponds to a thorax temperature of 47.9 $^{circ}$C.

direct measurement of the incident short-wave radiation (solar radiation), and $f$ is the fraction of the incident short-wave radiation that is reflected by the Earth (with default value $f$ = 0.25 for grassy surfaces [28]). Altogether, the heat gain from short-wave radiation is thus

$$Q_S = Q_{S_{in}} + Q_{S_{out}} = \epsilon_a a_{th} P(\alpha_p + \alpha_{np} f). \tag{2}$$

Heat gain due to long-wave radiation into the bee from the Earth is given by $Q_{L_e} = \alpha_{np} \epsilon_a a_{th} \sigma T_g^4$, and heat gain due to long-wave radiation into the bee from the atmosphere is given by $Q_{L_a} = \epsilon_a \delta T_{air}^6$, where $\alpha_{np}$ is the fraction of the surface exposed to the radiation, $a_{th}$

is the surface area of the bee's thorax, $\sigma$ is the Stefan-Boltzmann constant, $T_g$ is the temperature of the ground (soil temperature, default value 17.1 °C, typical for Ireland in the summer), $\delta$ is an experimentally determined parameter [29] that accounts for the effects of atmospheric emissivity, and $T_{air}$ is the air temperature. Altogether, the heat gain due to long-wave radiation is thus

$$Q_L = \alpha_{np}\epsilon_a a_{th}(\sigma T_g^4 + \delta T_{air}^6) \tag{3}$$

## Environmental Heat Loss Mechanisms

Heat loss via long-wave radiation from the bee is given by

$$Q_R = \epsilon_e a_{th} \sigma T_{th}^4 \tag{4}$$

where $\epsilon_e$ is the emissivity of the bee, $a_{th}$ is the surface area of the bee's thorax, $\sigma$ is the Stefan-Boltzmann constant, and $T_{th}$ is the thorax temperature.

Heat loss due to convection depends on the air temperature and the flight speed of the bee and is given by

$$Q_C = h a_{th}(s T_{th} - T_{air}) \tag{5}$$

where $a_{th}$ is the surface area of the thorax, $s$ is the reduction in bee surface temperature relative to core thorax temperature $T_{th}$, and $T_{air}$ is the air temperature. The heat transfer coefficient, $h$ is given by

$$h = \frac{c_d \kappa}{d_{th}} \left(\frac{v d_{th}}{\nu}\right)^n \tag{6}$$

where $c_d$ and $n$ are experimentally determined parameters, $\kappa = (0.02646 T_{air}^{1.5})/(T_{air} + 245.4 \cdot 10^{-12/T_{air}})$ W/m/K is the thermal conductivity of air [30], $\nu = (1.458 \cdot 10^{-6} \cdot T_{air}^{1.5})/(T_{air} + 110.4)$ m²/s is the kinematic viscosity of air [30], $d_{th}$ is the average thorax diameter (averaged from several sources in the literature), and $v$ is the flight speed.

## Physiological mechanisms

The basal or resting metabolic rate of an individual can be modelled as

$$Q_I = i_0 \left(\frac{m_b}{m_i}\right)^{3/4} e^{-\frac{E}{k}\left(\frac{1}{T_{th}} - \frac{1}{T_i}\right)}, \tag{7}$$

where $i_0$ is a normalization constant representing the metabolic rate (in W) for a bee with reference body mass $m_i$ at reference temperature $T_i$; $m_b$ is the mass of the modelled bee; $E = 0.63$ is the activation energy; $k$ is Boltzmann's constant; and $T_{th}$ is the thorax temperature (modified from [31] and [32]). Lacking any other model for the metabolic rate, we use the same functional form of metabolic rate with different parameter values for $i_0$ for flight and resting. This model for metabolic rate does not depend on the weather conditions.

Heinrich [26] studied honeybees under a variety test scenarios and found that head temperature (when below a certain upper limit) was primarily a function of thorax temperature, and remained at approximately $\Delta T_h = 2.9$ °C less than the thorax temperature over time. Sepúlveda-Rodríguez et al. [24] also found that heat is transferred from the thorax to the head during flight for bumblebees. Since bumblebees and honeybees have a similar body structure, we assume the head temperature is kept constant at $T_h = T_{th} - \Delta T_h$ for both bee groups. We

allow the head to experience heat gain and loss due to short-wave radiation, long-wave radiation, and convection. Thus, $Q_H = Q_{S_H} + Q_{L_H} - Q_{R_H} - Q_{C_H}$, where $Q_{S_H}$, $Q_{L_H}$, $Q_{R_H}$, and $Q_{C_H}$ are modelled the same as $Q_S$, $Q_L$, $Q_R$, and $Q_C$ (see above), replacing $T_{th}$ with $T_{th} - \Delta T_h$, and replacing the thorax surface area and diameter with the head surface area and diameter. Effectively, this means that the head is gaining sufficient heat from the thorax to keep it at a temperature $\Delta T_h = 2.9\,°C$ below the thorax temperature. As the head cools from environmental heat loss, it draws more heat from the thorax to maintain that temperature differential, cooling the thorax as well. Since $Q_H$ includes both heat gain and heat loss mechanisms, $Q_H$ will be negative when the head is losing heat to the environment. Therefore, the term is added to the model rather than being subtracted as the other heat loss mechanisms are.

The connection between the thorax and abdomen in bees is very narrow, creating a counter-current heat exchange that retains heat in the thorax (Fig 1 inset). Heat loss through the abdomen is therefore generally negligible unless a particular behavioural mechanism is employed (see below).

## Behavioural mechanisms

When bumblebees begin to overheat (thorax temperature $T_c = 42\,°C$), they modify their breathing and heartbeat patterns to interrupt the counter-current heat exchange and send heat to the abdomen [22]. The resulting heat loss depends on the difference in temperature between the thorax and the abdomen. Since we are not modelling the abdomen temperature directly, we simplify the model by keeping the abdomen temperature constant at air temperature, $T_{air}$, and do not model heat loss from the abdomen to the atmosphere as we do for the head and thorax. The cooling via the abdomen is

$$Q_A = \mathbb{1}_{ab} r_0 (T_{th} - T_{air}) \tag{8}$$

where $r_0$ is a normalising constant derived from [22] that represents the heat transfer rate per degree difference between the thorax and abdomen. For bumblebees, $\mathbb{1}_{ab}$ is either 1 or 0, allowing the cooling via the abdomen to be turned on or off depending on the thorax temperature. For honeybees, which do not use cooling via the abdomen, $\mathbb{1}_{ab} = 0$.

Honeybees are unable to interrupt the countercurrent heat exchange between the thorax and the abdomen [33]. Instead, when their head reaches a temperature of $T_c = 45\,°C$, they use a behaviour called 'tongue-lashing', in which they extend their tongue with a droplet of nectar, rapidly moving it in and out of the mouth. This cools the head via evaporative cooling of the nectar, which in turn cools the thorax as heat moves from the thorax to the head [17,26,27]. This is modelled as

$$Q_N = -\mathbb{1}_n \dot{m}_{evap} h_{fg} \tag{9}$$

where $\dot{m}_{evap}$ is the rate of evaporation and $h_{fg}$ is the latent heat of vaporization (of water). Details are provided in the Supporting Information. This evaporative cooling depends on the wind or flight speed, air temperature, and relative humidity. In order to simplify the interpretation of the results, we have set the wind speed to 0, so resting honeybees don't experience any evaporative cooling. For honeybees, $\mathbb{1}_n$ is 1 or 0 allowing the tongue-lashing behaviour to be turned on or off depending on the head temperature. For bumblebees, $\mathbb{1}_n = 0$ since they do not use tongue-lashing.

## Sensitivity analysis

We perform a full global sensitivity analysis, simultaneously varying all parameter values following the procedure outlined in [34]. We also perform a limited global sensitivity analysis, keeping the environmental parameters (ground reflectance, solar radiation, ground surface temperature, air temperature, latent heat of vaporisation of water, and relative humidity) constant at their default values and simultaneously varying all other parameters. Details are provided in Supporting information.

## Fitting $i_0$ and $E$

When we ran the model with the values for E suggested by the literature [31], it produced unrealistic results with respect to the expected temperature thresholds for flight. The limited sensitivity analysis (keeping environmental variables constant) revealed that the metabolic rate $i_0$, the flight speed, $v$, the mass of the bee $m_b$, and the activation energy $E$ are the top four variables accounting for the variability in the model results. Of these four parameters, $E$ and $i_0$ are fixed physiological characteristics, $v$ is a behaviour that can vary, and $m_b$ is a naturally varying trait. We therefore fit $i_0$ and $E$ to achieve as close to the expected temperature thresholds as possible, and explore the model results with respect to varying $v$ and $m_b$.

To fit $E$ and $i_0$, we use data on the air temperatures at which flight should be possible and at which cooling behaviours should begin ($T_{min}$ and $T_c$ respectively, Tables 1 and 2). We use a simple Approximate Bayesian Computation approach [35] in which we calculate the air temperature at which the bee would be able to fly and the air temperature at which the behavioural cooling should begin for each point on a grid of $E$ and $i_0$ values, repeatedly compare these values to randomly generated target air temperatures, selecting the closest one each time, and then take the values for $E$ and $i_0$ where the set of closest values has the highest density. Full details are provided in Supporting information.

## Model simulations

We used the fitted values of $i_0$ and $E$ along with all other default parameters, and solve equation 1 at various different air temperatures, flight speeds, and thorax masses to find the equilibrium thorax temperature in each case.

# Results

Individual solution curves to equation 1 show that in all cases the bee's thorax reaches an equilibrium temperature after some time (anywhere from 2–3 minutes to upwards of 30 minutes). This equilibrium temperature and the time it takes to reach it vary depending on which type of bee it is, what behaviour(s) the bee is engaging in, and what the air temperature is. A sample set of solution curves is shown in Fig 2.

## Heat exchange mechanisms

The majority of insect species produce very little heat metabolically, but bees are among those which produce large amounts of heat during flight and when activating their flight muscles for pre-flight warm-up. Including both resting and active metabolic rates (values for $i_0$) in our model gives rise to two distinct patterns of equilibrium thorax temperature in response to air temperature and is required to accurately capture the temperatures at which a bee will be able to fly (Fig 3). Including the heat transfer from the thorax to the head has an overall cooling effect, so the equilibrium thorax temperature is lower than it would be without the effect of the head.

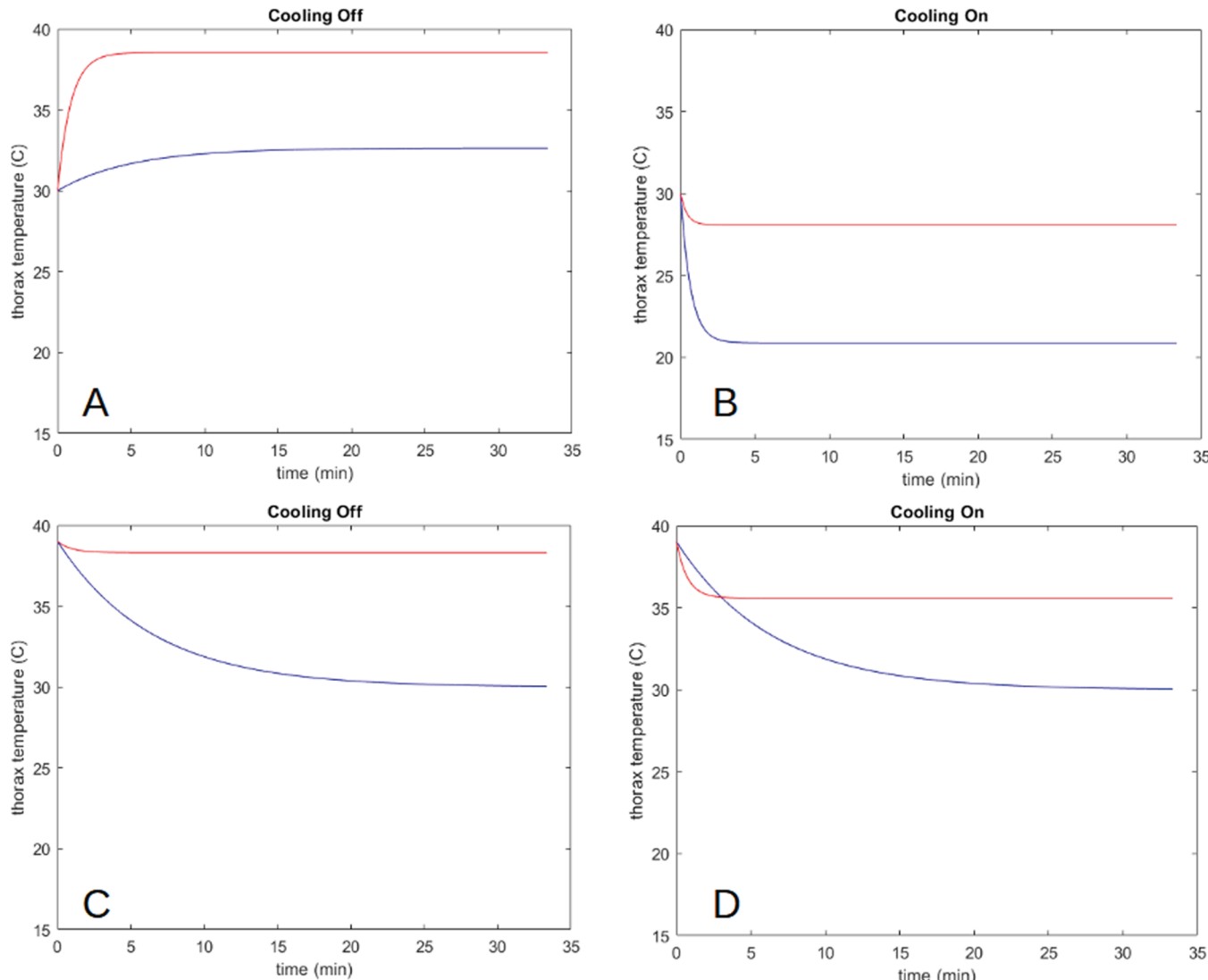

**Fig 2. Solution curves to the model.** (A) Bumblebee, without cooling behaviour. (B) Bumblebee, with cooling behaviour. (C) Honeybee, without cooling behaviour. (D) Honeybee, without cooling behaviour. $T_{air}$ = 20 °C and all other parameters at default/fitted values. Solutions for a resting metabolic rate are in blue and flying metabolic rate in red.

Behavioural cooling mechanisms allow bumblebees and honeybees to be active over a wide range of air temperatures. The linear trends seen in equilibrium thorax temperature as a function of air temperature both with and without the cooling mechanisms (Fig 3) extend across the full range of equilibrium thorax temperatures shown, and thus the bee would have much narrower thermal limits if these cooling mechanisms were either not possible or were always in effect. In our model, the evaporative cooling mechanism of honeybees is only effective when flying, so a resting honeybee is unable to further cool itself, whereas a resting bumblebee can cool through its abdomen.

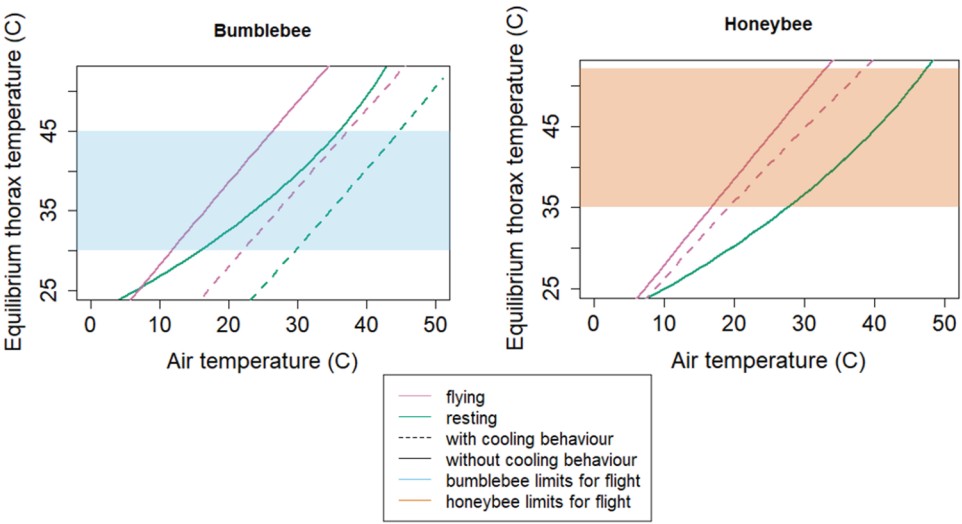

**Fig 3. Equilibrium thorax temperature versus air temperature.** Left: an average temperate bumblebee. Right: an average western honeybee. Pink curves indicate flight and green indicate resting. Solid lines indicate no behavioural cooling via the abdomen or tongue-lashing, and dashed lines indicate that behavioural cooling is employed. The shaded regions indicate the limits on thorax temperature for flight.

## Global sensitivity analysis

The full sensitivity analysis of the model indicates that, of the environmental parameters used in the model (ground reflectance, solar radiation, ground surface temperature, air temperature, latent heat of vaporisation of water, and relative humidity), air temperature explains the most variability in model output (46-55%), $P$ explains approximately 4–7%, and the others explain less than 1% each. We therefore present our results in terms of air temperature, and do not vary the other environmental parameters. According to our limited sensitivity analysis, when weather variables are disregarded, variation in the metabolic rate, $i_0$, accounts for almost half of the variation in thorax temperature. For both honeybees and bumblebees, flight speed, $v$, accounts for a significant portion of the remaining variability (48% and 38%, respectively). For bumblebees, mass of the bee, $m_b$, is also significant (14–19% depending on behaviour). The activation energy $E$, which effectively controls temperature dependence in the metabolic heat gain term, is the next most significant parameter in both cases (1–3%).

## Functional traits

Thorax mass and flight speed can vary significantly between individuals, and this has an effect on the upper thermal limit for sustained flight (Fig 4). Overall, smaller bees and faster flying bees are able to sustain flight at higher air temperatures. For honeybees, within their realistic range of masses, there is a wide range of possible maximum air temperatures depending on flight speed, but this range varies minimally as mass varies. Even when we expand the range of masses well beyond the realistic range, the mass has a minimal effect on the range of possible maximum air temperatures for sustained flight. For bumblebees, both smaller mass and faster flight speed result in a higher maximum air temperature when the other is held constant, but

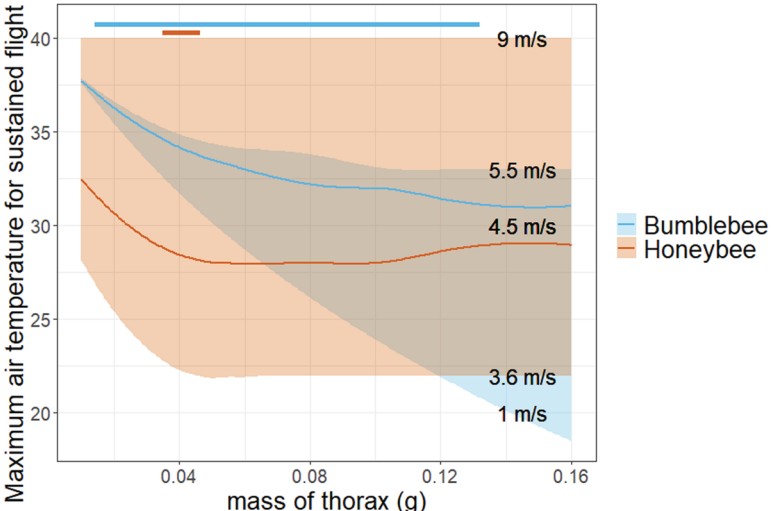

**Fig 4. The range upper thermal limits for sustained flight depending on its thorax mass and flight speed.** The realistic range for the thorax mass of each bee group is indicated by the horizontal bars at the top of the plot. Variability due to flight speed is indicated by the shaded area. The middle curve is a flight speed of 4.5 m/s in both cases. The increasing trend seen in honeybees greater than thorax mass 0.08 g (outside of the realistic range) is due to convective cooling having a proportionally greater effect at these masses.

there is also an interaction such that as mass decreases, the range of maximum air temperatures possible with varying flight speed decreases in size and increases in temperature. Bumblebees have a much wider realistic mass range than honeybees, but this relationship holds outside of that mass range as well.

## Discussion

Weather, and particularly air temperature, is an important determinant of bee activity and is likely to change with changing climate, potentially changing the delivery of pollination services. To predict the effects of weather, and specifically air temperature, solar radiation, relative humidity, and ground temperature, on bee activity, we have developed a mechanistic model of thermoregulation in insects and applied the approach to two key bee groups with different functional traits—bumblebees occurring in temperate regions and the western honeybee. We found that behaviourally modulated cooling mechanisms (i.e., allowing excess heat to pass from the thorax into the abdomen or controlling evaporative cooling by extending the tongue with a droplet of nectar during flight), in combination with the well known mechanism of shivering to warm up before flight, are necessary for bees to remain active across a wide range of air temperatures. Using our default parameters, representing an average honeybee and an average bumblebee, and both warming and cooling behaviours, bumblebees can sustain flight at air temperatures of approximately 11–36 °C and honeybees over a range of approximately 16–38 °C (Fig 3, lower equilibrium temperature within the thorax temperature range on the pink solid line and upper temperature on the pink dashed line). Our model also reflects the well known fact that bumblebees and honeybees are able to warm themselves before flight. Without the ability to warm before flight a bumblebee's thorax would be only warm enough for flight at an air temperature of approximately 15 °C and a honeybee's only at approximately 28 °C, and without the ability to engage cooling behaviour, the upper limits on air temperature for flight would be 26 °C and 32 °C respectively (Fig 3, solid lines only). With

constant cooling, but including warming before flight, the air temperature ranges for flight would be 21–36 °C for bumblebees and 19–38 °C for honeybees respectively (Fig 3, dashed lines only).

When we vary the bees' thorax mass and flight speed, the maximum temperature at which sustained flight is possible varies. For bumblebees, which naturally have a large variation in mass, thorax mass and flight speed together affect the maximum air temperature for sustained flight, anywhere from 21 °C for the highest realistic thorax mass and lowest flight speed to approximately 36 °C for the lowest realistic thorax mass and highest flight speed. For honeybees, which naturally have very little variation in mass, flight speed affects this thermal limit, ranging between approximately 22 °C and 40 °C across all realistic thorax masses, but thorax mass has little impact. Outside the realistic range of honeybee thorax masses, only for the very smallest bees is there any great effect of thorax mass on the maximum temperature for sustained flight. For (hypothetical) honeybees with a thorax mass between 0.01 g and 0.04 g, smaller bees have a higher maximum temperature at which they could sustain flight.

Overall, our model predicts that honeybees have a higher maximum thermal limit for sustained flight than bumblebees, supporting existing empirical knowledge. The higher maximum thermal limit of honeybees appears to be primarily due to the fact that honeybees are generally faster flyers than bumblebees and have a wider range of flight speeds. Since evaporative cooling depends on flight speed, the higher maximum flight speed of honeybees gives them a greater maximum thermal limit. However, between a bumblebee and honeybee of the same mass, both flying at 4.5 m/s, the bumblebee would have a higher maximum thermal limit (Fig 4, solid lines). This suggest that convective cooling is more effective in bumblebees than honeybees, and is in fact more effective than convective cooling and evaporative cooling combined in honeybees, as convective cooling is the only mechanism where flight speed is a factor in the bumblebee paramaterization of the model.

We also predict that a resting bumblebee engaged in cooling behaviour via the abdomen reaches a lethal thorax temperature ($CT_{max}$) at an air temperature of about 42 °C, whereas a resting honeybee, whether or not it is behaviourally cooling, reaches the lethal thorax temperature at about 47 °C. The differences in these thermal limits could explain why bumblebees are rare in tropical regions, and suggest that in temperate regions where bumblebees have the highest species diversity there could be more scope for pollination by honeybees than by bumblebees as the global climate warms. However, we note that one study found that lab reared *Bombus impatiens* reached $CT_{max}$ (thorax temperature 42–44 °C) at ambient temperatures of 52–55 °C [39], and another found that commercially reared *Bombus terrestris audax* reached $CT_{max}$ (determined by behaviour) at ambient temperatures of 48.9–52.7 °C [46]. The difference between these results and our model results could be due to differences in laboratory conditions and our modelled field conditions, or indicate that there are cooling mechanisms not included in our model that allow bumblebees to maintain their thorax temperature below $CT_{max}$ at higher ambient temperatures. Either way, it appears that our model provides a conservative estimate of the maximum air temperatures that bumblebees can withstand.

Geographical range shifts to higher elevations and losses at the southern end of ranges have been observed in bumblebees, and are predicted to continue as climate warms [47,48]. This is consistent with lower thermal limits for bumblebees that our model predicts. As temperatures have risen and continue to do so, bumblebees would need to move northwards or upwards to stay within their thermal limits, and might also be unable to continue foraging in more southern regions. Alternately, they would need to adapt their functional traits. Thus, honeybees may become increasingly important as the climate warms, and plants that rely on bumblebee pollinators may suffer.

We gain insights into differences between honeybees and bumblebees in the mechanisms controlling their thermal limits. The honeybee's primary behavioural cooling mechanism of the evaporation of nectar droplets from its extended tongue requires air to flow over the bee. The cooling effect increases as the air speed increases, so it is dependent on flight speed when bee is in flight. While it is not known if honeybees deliberately fly faster to cool off, they do display broader range of flight speeds than bumblebees. Conversely, a resting honeybee's ability to actively decrease its head, and therefore thorax, temperature is at the mercy of the ambient wind speed. For bumblebees, the primary behavioural cooling mechanism (heat transferred from the thorax to abdomen) as we have modelled it is independent of flight or wind speed, so thorax temperature can be controlled equally well when flying or resting. However, it may be that we have overestimated the cooling effects of heat transfer to the abdomen. Sepúlveda-Rodríguez et al. [24] found that immediately post-flight, bumblebees had an abdomen temperature cooler than the thorax temperature but warmer than the air temperature. If abdomen temperature is warmer than air temperature instead of the same as air temperature as we have modelled it, then less heat can be transferred to it from the thorax and the wind speed will also affect how quickly the abdomen can cool. Although our model suggests that bumblebees may have greater cooling capacity than honeybees, especially on windless days, this might not be the case. A more complex model which also models the abdomen temperature directly could be used to explore this topic.

At an air temperature of 40 °C, and assuming no wind, our model predicts that an average bumblebee could, through resting while activating cooling behaviour, maintain a thorax temperature close to air temperature (i.e. 40 °C) whereas the thorax temperature of a resting honeybee, which cannot cool itself while resting in our model, would be approximately 45 °C. We therefore hypothesize that a cooling behaviour which is effective when the bee is not in flight could allow bees to effectively cool themselves by stopping to rest mid-foraging, and therefore continue providing pollination services even at higher temperatures, albeit at a reduced level. We also see the effect of bumblebees' overall higher metabolic rate in their ability to begin flying at cooler temperatures than honeybees, consistent with empirical observations. For both bee groups, the ability to warm up before flight will remain an important aspect of their ability to provide pollination services over a broad range of temperatures, even as we see global temperatures rising due to climate change.

Our model shows that thorax mass is an important functional trait in bumblebees from temperate regions, but not honeybees (specifically *Apis mellifera*). Although the range of masses that naturally occur in *Apis mellifera* is small, using a mechanistic model allows us to extrapolate beyond that range more readily than a statistical model would, and so allows us to make inferences about *Apis mellifera* outside of the currently observed range of thorax masses. Bumblebees with smaller thorax masses can be active in hotter conditions, which may produce evolutionary pressure for smaller bumblebees in temperate regions as global mean temperatures rise. Evolutionary shifts towards smaller masses have already been observed in species occurring in temperate regions[47], and our model supports the hypothesis that this is due to climate change [49]. Body mass has consequences for foraging ability beyond activity level: larger bumblebees can have a greater foraging rate and overall greater foraging success [50], and have been shown to deposit more pollen per flower visit [51] as well as producing buzzes of greater amplitude, thereby releasing more pollen [52]. If bumblebees decrease in size in response to climate change, we might then expect to see corresponding decreases in the delivery of pollination services. However, this is not the case for honeybees, where thorax mass has relatively little effect on the thermal limits. We would therefore not expect to see an evolutionary shift in honeybee masses, or a change in pollination services of these types.

Changing weather patterns due to global climate change are not the only stressors that bees face. Agricultural land use, and particularly monocultures, put bees at risk of nutritional deficits and increase pesticide exposure. The effects of nutrition and pesticide exposure specifically on bees' ability to thermoregulate have been studied very little. However, in one study, exposure to the insecticide flupyradifurone in combination with a diet of nectar with a low sugar concentration was found to decrease the ability of *Apis mellifera* L. to thermoregulate [53]. Another study found that nutritional limitations had a short term effect on the thermal tolerance of *Bombus impatiens* [54]. The effects of both pesticide exposure and nutritional deficits on flight thermoregulation are promising avenues of future lab and field research, which could be used to refine and improve the predictions of this or other similar models.

The mechanisms of metabolic heat production ($Q_I$) and evaporative heat loss ($Q_N$) have some uncertainty in their formulation. Bees are relatively unique among insects in the quantity of heat they produce metabolically, making this model an important addition to the small body of other insect thermoregulation and heat budget models, which have previously focused on the environmental mechanisms of heat exchange (e.g. [16,18,55,56]). There is, however, considerable uncertainty in the general scientific community about the form that the rate of metabolic heat production should take, particularly for active, as opposed to resting, rates (e.g. [57–59]). This uncertainty is reflected in our model. Our fitted value for *E* for both honeybees and bumblebees is substantially lower than the literature value, effectively reducing the temperature dependence of metabolic heat production. It is therefore possible that the model for active metabolic rate should take a completely different form than the resting metabolic rate model from [31] that we have used here. Experimental studies are needed to inform this aspect of our model. Additionally, evaporative cooling is a very complex process, and the model we have chosen is a relatively simple one. Follow on work should compare our simple model with more complex approaches and look at the model's sensitivity to these conceptual changes. Other possible mechanisms of heat flux that could be included to improve the model are passive evaporative heat loss [17] and small amounts of heat loss through the abdomen when the counter-current heat exchange is in effect [60].

Given the large amount of research on the western honeybee *Apis mellifera*, most parameters used for this species in this model are from this single species. However, as bumblebees are diverse (approximately 250 species) and different species have been domesticated in different parts of the world (e.g. *Bombus terrestris* in Europe and *Bombus impatiens* in North America), it wasn't possible to find values for all parameters from a single species. Therefore our values have come from a variety of different species. Although the mechanistic structure of the model makes inferences outside of the observed information possible, nonetheless our mixture of parameters may not represent all bumblebee species equally well. The parameters could be updated for individual species as more research emerges in the future.

We have applied our model to two bee groups (temperate bumblebees and the western honeybee), but it could be extended to other bee and insect species. Environmental heat exchange mechanisms apply widely across insect species, moderated by differences in body shape, size, and surface characteristics (e.g. bees with hair versus beetles with reflective casings). However, metabolic heat production and behavioural thermoregulation strategies can vary widely between different species [61]. Insect species that would fit within the current model include blowflies, which cool their heads in a similar manner to honeybees [62], and carpenter bees and large morph male Sonoran desert bees (*Centris caesalpiniae*), which have also been observed to control heat transfer from the thorax to the abdomen [25,63]. Some species of solitary bees do not appear to thermoregulate either by warming themselves metabolically before flight or by using behavioural cooling mechanisms the way bumblebees and honeybees do, but instead bask and take advantage of microclimates inside of flowers [64]

or fly selectively based on ambient temperatures [65]. These differences would not necessarily require additional mechanisms in the model, but certain mechanisms might need to be applied differently (e.g. radiative heat gain at basking and non-basking levels). Other insect species would require additional mechanisms added to the model. For example, some species of dragonflies control heat transfer to the head [66], and another species has a specialized wing surface that increases cooling [67]. These mechanisms could be added to our model if required and the same analysis approach used to obtain analogous thermal limits.

In summary, we have developed a novel mechanistic model for insect thorax temperatures that can be used to determine the effects of all heat exchange mechanisms and thermoregulatory behaviours on the equilibrium thorax temperature of an insect in flight or at rest. We specifically used this model to calculate the thermal limits for western honeybees and temperate bumblebees, and to show that they will respond differently to weather effects, and therefore potentially to climate change, due to differences in their cooling behaviours and functional traits. As these honeybees are largely managed in areas where both groups co-exist, and the bumblebees are largely wild, we suggest that having mixed pollinator management strategies using managed species while also promoting wild populations of bees may have the best outcome in terms of providing resilience of crop pollination to future climate change.

## Supporting information

**S1 Appendix. Supporting information.**
(PDF)

## Author contributions

**Conceptualization:** Sarah MacQueen, Dara A. Stanley, Jon M. Yearsley.

**Data curation:** Sarah MacQueen.

**Formal analysis:** Sarah MacQueen.

**Funding acquisition:** Dara A. Stanley.

**Investigation:** Sarah MacQueen.

**Methodology:** Sarah MacQueen, Jon M. Yearsley.

**Project administration:** Dara A. Stanley.

**Resources:** Dara A. Stanley.

**Supervision:** Dara A. Stanley, Jon M. Yearsley.

**Validation:** Sarah MacQueen, Jon M. Yearsley.

**Visualization:** Sarah MacQueen.

**Writing – original draft:** Sarah MacQueen.

**Writing – review & editing:** Sarah MacQueen, Dara A. Stanley, Jon M. Yearsley.

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
