## [Decision Letter · Decision Letter 0]

10 Apr 2024

PONE-D-23-42752Thermal limits of bumblebees and honeybees are modulated by different functional traits: predictions of a mechanistic modelPLOS ONE

Dear Dr. MacQueen,

Thank you for submitting your manuscript to PLOS ONE. After careful consideration, we feel that it has merit but does not fully meet PLOS ONE’s publication criteria as it currently stands. Therefore, we invite you to submit a revised version of the manuscript that addresses the points raised during the review process.

**In particular, both reviewers raise issues with the lack of clarity about parts of the model. It would be important to address these issues and provide further explanations. It would also be good to take note  of the required qualifications to statements in the text with reference to previous work suggested.**

We look forward to receiving your revised manuscript.

Kind regards,

Vivek Nityananda

Academic Editor

PLOS ONE

Journal Requirements:

Reviewers' comments:

Reviewer's Responses to Questions

**Comments to the Author**

1. Is the manuscript technically sound, and do the data support the conclusions?

Reviewer #1: Yes

Reviewer #2: Partly

2. Has the statistical analysis been performed appropriately and rigorously? 

Reviewer #1: Yes

Reviewer #2: N/A

3. Have the authors made all data underlying the findings in their manuscript fully available?

Reviewer #1: Yes

Reviewer #2: No

4. Is the manuscript presented in an intelligible fashion and written in standard English?

Reviewer #1: Yes

Reviewer #2: Yes

5. Review Comments to the Author

Reviewer #1: In this manuscript, the authors present a thermodynamic model for insect thorax temperatures that includes metabolic processes and behavioural adaptations. The study is interesting, thorough and paves the way for species specific models on adaptation to changing climatic conditions while considering specific behavioural adaptations. I found the manuscript easy to read and understand for the most part but have some major and minor comments:

Major comments:

1. Questions on Qh: I found the explanation of how Qh is calculated difficult to understand. From Fig. 1, Qh is implied as heat taken away from thorax. In Eq. 1, Qh is added to (and not subtracted from) the thoracic temperature. In the text (L104-109) Qh is a combination of heat gain and loss due to environmental processes similar to what is described for the thorax. However, for honey bees the authors point out that the temperature differential between head and thorax is maintained at a constant value irrespective of this heat gain and loss. I think the authors should improve their description of Qh to make it’s definition and role clearer.

2. Several assumptions of parameter values have been made in the model without clear explanations. For example, one assumption in the model related to Qh is that the temperature differential between head and thorax is the same for bumblebees as it is for honey bees. This assumption is not present in the main text but is marked in the supplementary table 4. This and other assumptions should be explicitly stated and justified in the Methods section and should not be delegated only to the supplementary information as it is essential to understand the model.

3. Discussion regarding the effect of body mass: I found some of the inferences and discussion on this topic circular. In the model inputs, the body mass of bumblebees and honey bees are input with different levels of variation (for bumblebees the range is nearly 10 fold while the variation in honey bees is much less). While I understand that this corresponds to the extant variation seen, this makes it difficult to disentangle whether the effects of body mass on flight in the model is simply due to the reduced variation. One way would be to model similar levels of variation around the mean body mass in honey bees and bumblebees. While artificial, this will help strengthen the authors’ claim of body mass being an important factor for bumblebees but not honey bees.

4. Behavioural cooling mechanisms: This was an interesting addition to the model. However, I have some questions. Why is active cooling through evaporation of nectar only possible during flight in honey bees? The discussion states that it is strongest during flight (L220) but in the model it is taken to be completely absent (and not weaker) during rest. Can the authors clarify why they chose to not include it as a behavioural process in resting honey bees?

5. Fig.2: I think this figure could be expanded by adding panels from Supplementary figure 1. The results on temperature ranges for flight with and without cooling are an important part of the manuscript (it is highlighted in the first paragraph of the discussion, L189-194) and it is not clear why they are not presented in the main results.

6. L143-145: I had difficulty understanding how and why the parameter E was chosen for the restricted sensitivity analysis, given that it only explains very little amount of the variation (supplementary tables 3 and 4). E is only present when non-environmental variables are considered and even then, it only accounts for 1-3% of the remaining variation. The authors should justify their choice of using E more clearly.

7. In the parameters which require input of surface area values, only the surface area of the thorax or the head is provided in the various equations. Is heat gain or loss through the abdomen negligible? Can the authors elaborate on this.

Minor comments:

8. The introduction was well written and covered the major points succinctly. I enjoyed reading this section.

9. L64: It is not clear to me why certain subscripts in Eq 1 are in small letters (Qab) compared to others? What is the convention being used?

10. L96: How is the average thorax diameter calculated?

11. L108: the section number is missing.

12. L210: Bumblebees is wrongly spelt as 'bumblebeest'

13. L229-230: I’m not sure how cooling capacity benefits bumblebees more mid-foraging given that in both species the active cooling process is present during flight.

14. L262 ‘improce’ should be replaced with ‘improve’.

Reviewer #2: This paper presents a mechanistic model for predicting the thermal limits of activity in honeybees and bumblebees to facilitate predictions of how future climate changes will affect crop pollination. The model is based primarily on empirical data but is designed to be applied beyond these limits. In general, the text is well-written but is lacking important clarifications and includes generalisations and implicit assumptions that make the results of the model difficult to interpret and would make it problematic for other research groups to apply. As such, the work is not suitable for publication in its present form but could be if the concerns below are adequately addressed.

General comments

While the authors’ attempt to develop a general model by including two groups of bees is commendable, the work is limited by the lack of empirical data from more than a few species (and only one for honeybees). The general oversight regarding potential differences between honeybee and bumblebee species that inhabit different environments is the greatest weakness of the work. Although it is understandable that the authors are necessarily limited because the empirically measured parameters required for the model are only available for a limited number of species, it is important for this limitation to be made explicit. For example, bumblebee species (of which there are approximately 250) occupy a broad range of habitats, including the Arctic, the Mediterranean, the deserts of North America and the Amazon rainforest. Honeybee species also vary in both their habitat and their body size, which would also potentially affect the generalisability of the results. Because of this, it is unlikely that the parameters used for the model apply to species that have evolved in these more extreme environments and it is therefore important that the authors are aware of this and discuss this limitation in the text. The other problem associated with the limited number of species used in the model is that the conclusion that honeybee body size does not affect their thermal limits may be incorrect. This is because the honeybee species from which all empirical data was made tends to have workers that do not vary in body size. However, different honeybee species have very different body sizes and this could result in differences in thermal limits. As the authors do not clarify that the honeybee species they refer to is Apis mellifera, this leads to the impression that these results would apply to other honeybee species with different body sizes, but this may not be the case.

Line 31, 55 and line 181: the model apparently only accounts for temperature changes and not changes in other weather factors, such as cloud cover, wind or humidity. If this is not the case, then please specify what other weather variables were used, if this is the case then clarify by replacing ‘weather’ here with ’temperature’.

Equation 1: define the terms dt and dTth in the text before presenting the model

Equation 1: please also explain why QH is being added rather than subtracted here. Potential heat transfer from the thorax to the head would reduce the thorax temperature rather than increase it as suggested in this equation. Please clarify the reasoning behind this in the text.

Line 68: This statement should be supported with an explanation or a citation “and head transfer from the thorax to the head are necessary for bees”. There appears to be only two published studies that explore heat transfer to the head in bees and it appears as though they can regulate this. These works should be cited and the statement revised to reflect the state of our knowledge on this:

Meredith G. Johnson, Jordan R. Glass, Jon F. Harrison; A desert bee thermoregulates with an abdominal convector during flight. J Exp Biol 1 October 2022; 225 (19): jeb244147. doi: https://doi.org/10.1242/jeb.244147

Guadalupe Sepúlveda-Rodríguez, Kevin T. Roberts, Priscila Araújo, Philipp Lehmann, Emily Baird, Bumblebee thermoregulation at increasing temperatures is affected by behavioral state, Journal of Thermal Biology, 2024, 103830,

https://doi.org/10.1016/j.jtherbio.2024.103830.

Line 74: Isn’t the reflection of shortwave radiation from the Earth dependent on the substrate (i.e. if it is long grass, short grass, mud, water etc.)? If so, please explain the assumption of reflectance being made for this equation and note any potential caveats about substrate in the text.

Equation 2: define �P in the text

Equations 2 and 3: These equations are quite confusing. Why is the surface area of the bee’s thorax specifically important for the calculations of Qs and QL and why is it treated separately from the body surface calculations? Please clarify in the text.

Line 84: Where does this measure of ground temperature come from? Please explain in the text.

Line 103: This is quite confusing. The resting metabolic rate was used for the model but then applied to flying as well but these are two very different values in most insects. Furthermore, the metabolic rate for flight varies with air temperature. Please clarify how using the resting metabolic rate value is valid.

Line 104: The basis for this assumption is unclear. Why would a connection between the head and thorax affect the heat flux from the head to the environment? Would active transfer of warm haemolymph from the thorax to the head (as appears to be the case in bumblebees) affect this assumption? If so, how?

Line 105: This statement about a constant difference in head temperature was based on studies from honeybees and does not appear to be true for bumblebees or desert bees. The model should be revised to reflect this

Line 108: the reference to a section (in parentheses) is incomplete.

Line 116: The paper cited here is a study on honeybees and not on bumblebees, this value would also depend on whether the bees are resting or flying. Please clarify for the reader what conditions this statement is specific for.

Line 117: the conclusion that “flight speed thus plays no role in abdomen cooling” should be more clearly supported, as convective cooling will increase with flight speed and should therefore affect the abdomen cooling.

Table 1: Please specify the species for which these values were taken as it is important for the model and for the reader to know how specific they are. Particularly for bumblebees, there are around 250 different species that inhabit a broad range of habitats and that have different body sizes and hairiness. They likely have differences in their ability to thermoregulate and in their responses to temperature. The model likely uses data from very few, if more than one, species of honeybee and bumblebee but the authors do not highlight this as a limitation. Please address this in the Table and in the text, discussing how parameters taken from one, or just a few species might skew the results of the model.

Line 163: It is not clear what other environmental parameters the authors are referring to. Please clarify what they are here.

Line 174: when referring to ‘…ones experiencing greater wind speeds’, I assume that the authors mean head winds or would winds from other directions also have this effect? Please clarify in the text.

Line 189: The term ‘indefinitely’ is used incorrectly here and is misleading. Please revise.

Line 191: This conclusion does not make sense as how could bumblebees fly at 15°C if they cannot warm up their flight muscles? It is also an irrelevant statement as bumblebees do have the ability to warm up their flight muscles and to transfer heat to the abdomen during flight.

Line 201: correct the spelling of bumblebees

Line 205: Please provide a further clarification of why bumblebees would have a higher thermal limit.

Line 209: this statement is incorrect. Some bumblebee species do inhabit tropical environments, Bombus transversalis lives in the tropical lowlands of the Amazon rainforest, for example. Other bumblebee species live in deserts so they are not limited only to cold climates as implied here. Also, the honeybee Apis mellifera evolved in the tropical rainforests of India, so it is of course going to be adapted to flight in tropical environments. Please modify this statement to make it valid.

Line 223: This statement is incorrect as the primary behavioural cooling mechanism is dependent on flight speed as noted above (unless the authors can provide empirical data to show otherwise). Please modify.

Line 232: the use of the term ‘species’ here is incorrect as the terms honeybee and bumblebee are ambiguous and could refer to genera or to specific species. Please be more explicit with the terminology and what is meant by it in the text to avoid confusion.

Line 235 (and lines 244-255): As the empirical data used for honeybees in the model comes from one species, Apis mellifera, this statement is not correct without specification. Honeybee workers are not size polymorphic as bumblebee workers are, so the empirical data is likely only taken from bees that vary little in body size. This lack of size variation for honeybees and not bumblebees could have a strong effect on the conclusions made in the model and the authors should be explicit about this in the text and revise this statement to clarify.

Line 236: The information in this sentence is not supported by evolution. As mentioned above, bumblebee species do live in tropical environments and they do not have smaller body sizes than those that occupy temperate climates. The authors either need to specify what species they are referring to or to include more correct information that accounts for species differences.

6. PLOS authors have the option to publish the peer review history of their article (what does this mean?). If published, this will include your full peer review and any attached files.

Reviewer #1: No

Reviewer #2: No

---

## [Author Response · Author response to Decision Letter 1]

19 Nov 2024

Reviewer #1: In this manuscript, the authors present a thermodynamic model for insect thorax temperatures that includes metabolic processes and behavioural adaptations. The study is interesting, thorough and paves the way for species specific models on adaptation to changing climatic conditions while considering specific behavioural adaptations. I found the manuscript easy to read and understand for the most part but have some major and minor comments:

Major comments:

1. Questions on Qh: I found the explanation of how Qh is calculated difficult to understand. From Fig. 1, Qh is implied as heat taken away from thorax. In Eq. 1, Qh is added to (and not subtracted from) the thoracic temperature. In the text (L104-109) Qh is a combination of heat gain and loss due to environmental processes similar to what is described for the thorax. However, for honey bees the authors point out that the temperature differential between head and thorax is maintained at a constant value irrespective of this heat gain and loss. I think the authors should improve their description of Qh to make it’s definition and role clearer.

Because Q_H itself is calculated by adding environmental heat gain and subtracting environmental heat loss, it will be negative when the head is losing heat. Thus, we have to add it to the model in order for it to have the effect of subtracting heat. Heat transfer from the thorax to the head was found to be primarily due to passive conduction (Heinrich 1980a), so maintaining T_h = T_th- ΔT_h effectively models this passive heat transfer from the thorax to the head. As the head cools due to environmental heat transfer, it draws more heat from the thorax to maintain that temperature differential, cooling the thorax as well.

We have designed the model in this way to avoid having to either solve for T_h or to model the rate of heat transfer from the thorax to the head, both of which would require additional assumptions (and/or empirical data that we are lacking) and increase the complexity of the model and analysis. We have added text clarifying these points when the model is introduced (lines 70-73) and in the section on physiological heat flux (lines 131-136).

2. Several assumptions of parameter values have been made in the model without clear explanations. For example, one assumption in the model related to Qh is that the temperature differential between head and thorax is the same for bumblebees as it is for honey bees. This assumption is not present in the main text but is marked in the supplementary table 4. This and other assumptions should be explicitly stated and justified in the Methods section and should not be delegated only to the supplementary information as it is essential to understand the model.

Thanks for pointing this out. We made the assumption about the differential between head and thorax temperature for bumblebees lacking any empirical evidence one way or the other. (See response to point 1 for further justification.) We have also clarified that in general, whenever information about either honeybees or bumblebees is lacking, we use the information we have for the other bee group.

3. Discussion regarding the effect of body mass: I found some of the inferences and discussion on this topic circular. In the model inputs, the body mass of bumblebees and honey bees are input with different levels of variation (for bumblebees the range is nearly 10 fold while the variation in honey bees is much less). While I understand that this corresponds to the extant variation seen, this makes it difficult to disentangle whether the effects of body mass on flight in the model is simply due to the reduced variation. One way would be to model similar levels of variation around the mean body mass in honey bees and bumblebees. While artificial, this will help strengthen the authors’ claim of body mass being an important factor for bumblebees but not honey bees.

We addressed this in Figure 3, where we simulated both bumblebees and honeybees across a range of thorax masses from 0.01g to 0.16g. This range is wider than the realistic range for both honeybees and bumblebees. The patterns we describe are observed across this entire (unrealistic) range. We have added text to the results (lines 236-247) and discussion (lines 346-350) to clarify this point.

4. Behavioural cooling mechanisms: This was an interesting addition to the model. However, I have some questions. Why is active cooling through evaporation of nectar only possible during flight in honey bees? The discussion states that it is strongest during flight (L220) but in the model it is taken to be completely absent (and not weaker) during rest. Can the authors clarify why they chose to not include it as a behavioural process in resting honey bees?

Any movement of air over the bee would give rise to evaporative cooling, so the underlying structure of the model allows for evaporative cooling when the bee isn’t flying. However, to simplify the model, we decided to keep wind speed to zero. Adding in wind would complicate the interpretation of the results, as we would have to include multiple wind speeds, and decide how many and what wind speeds to consider, as well as any directional effects while the bee is flying.

We have added text to clarify these points in the methods (lines 162-164) section, and modified the discussion (lines 315-319) to reflect the possible effects of wind during resting behaviour.

5. Fig.2: I think this figure could be expanded by adding panels from Supplementary figure 1. The results on temperature ranges for flight with and without cooling are an important part of the manuscript (it is highlighted in the first paragraph of the discussion, L189-194) and it is not clear why they are not presented in the main results.

We originally thought that figure 2 (now figure 3) as presented in the original manuscript gave a clearer picture, but upon reflection we agree with the reviewer that the figure as presented in the supplementary material is better. We have updated the text in the discussion (lines 263-272) to better reflect the new figure.

Although we believe the reviewer was referring to Supplementary figure 2 here (which corresponds to what was originally figure 2 in the main manuscript) we think that Supplementary figure 1 also enhances the main manuscript, and have therefore included it To accommodate this, we added a short section to the methods (Model Simulations, lines 193-196), a description of the results at the beginning of the results section (lines 198-201), and a new figure, now figure 2.

6. L143-145: I had difficulty understanding how and why the parameter E was chosen for the restricted sensitivity analysis, given that it only explains very little amount of the variation (supplementary tables 3 and 4). E is only present when non-environmental variables are considered and even then, it only accounts for 1-3% of the remaining variation. The authors should justify their choice of using E more clearly.

Thank you for pointing out this lack of clarity. When we ran the model with the values for E suggested by the literature (Brown et. al, 2004), it produced unrealistic results with respect to the expected temperature thresholds for flight. The limited sensitivity analysis (keeping environmental variables constant) revealed that the metabolic rate i_0, the flight speed, v, the mass of the bee m_b, and the activation energy E are the top four variables accounting for the variability in the model results. Of these four parameters, E and i_0 are fixed physiological characteristics, v is a behaviour that can vary, and m_b is a naturally varying trait. We therefore fit i_0 and E to achieve as close to the expected temperature thresholds as possible, and explore the model results with respect to varying v and m_b. We have updated the methods section (lines 175-178) with this information.

7. In the parameters which require input of surface area values, only the surface area of the thorax or the head is provided in the various equations. Is heat gain or loss through the abdomen negligible? Can the authors elaborate on this.

Due to the very narrow connection between the thorax and abdomen, heat loss from the thorax to the abdomen is negligible, and we therefore disregard the abdomen except in the case of behavioural cooling by bumblebees. When behavioural cooling using the abdomen is activated, we simplify the model by considering the abdomen to be a heat sink and its temperature to be the same as the ambient air temperature. That is, heat flows from the thorax to the abdomen at a given rate, and then is disregarded..

To clarify this point, we have moved the text explaining the counter-current heat exchange from the Behavioural Mechanisms subsection to the Physiological Mechanisms subsection (lines 137-140) and updated the description of Q_A (lines 142-148). We have also added some text to the beginning of the methods section explaining that we are only directly modelling the thorax temperature, and how the head and abdomen factor into this (lines 50-55).

Minor comments:

8. The introduction was well written and covered the major points succinctly. I enjoyed reading this section.

Thank you!

9. L64: It is not clear to me why certain subscripts in Eq 1 are in small letters (Qab) compared to others? What is the convention being used?

The convention being used is human error. We have no idea why “ab” and “n” were in lower case. We have changed Q_ab to Q_A to be consistent with the other one letter capital subscripts and Q_n to Q_N for consistency with the capitalization.

10. L96: How is the average thorax diameter calculated?

We took the average value from a few different sources in the literature. We have added text to this effect in the manuscript (Lines 108-109).

11. L108: the section number is missing.

Fixed

12. L210: Bumblebees is wrongly spelt as 'bumblebeest'

Oops, thanks!

13. L229-230: I’m not sure how cooling capacity benefits bumblebees more mid-foraging given that in both species the active cooling process is present during flight.

Cooling capacity could benefit bumblebees mid-foraging because they could stop to rest and keep cooling via the abdomen while resting, unlike honeybees. We have updated the text to include this information (lines 336-337). In the course of responding to reviewer 2’s comments we have also shifted this discussion away from bumblebees vs. honeybees and focussed on the types of cooling mechanism.

14. L262 ‘improce’ should be replaced with ‘improve’.

Oops, thanks!

Reviewer #2: This paper presents a mechanistic model for predicting the thermal limits of activity in honeybees and bumblebees to facilitate predictions of how future climate changes will affect crop pollination. The model is based primarily on empirical data but is designed to be applied beyond these limits. In general, the text is well-written but is lacking important clarifications and includes generalisations and implicit assumptions that make the results of the model difficult to interpret and would make it problematic for other research groups to apply. As such, the work is not suitable for publication in its present form but could be if the concerns below are adequately addressed.

General comments

While the authors’ attempt to develop a general model by including two groups of bees is commendable, the work is limited by the lack of empirical data from more than a few species (and only one for honeybees). The general oversight regarding potential differences between honeybee and bumblebee species that inhabit different environments is the greatest weakness of the work. Although it is understandable that the authors are necessarily limited because the empirically measured parameters required for the model are only available for a limited number of species, it is important for this limitation to be made explicit. For example, bumblebee species (of which there are approximately 250) occupy a broad range of habitats, including the Arctic, the Mediterranean, the deserts of North America and the Amazon rainforest. Honeybee species also vary in both their habitat and their body size, which would also potentially affect the generalisability of the results. Because of this, it is unlikely that the parameters used for the model apply to species that have evolved in these more extreme environments and it is therefore important that the authors are aware of this and discuss this limitation in the text. The other problem associated with the limited number of species used in the model is that the conclusion that honeybee body size does not affect their thermal limits may be incorrect. This is because the honeybee species from which all empirical data was made tends to have workers that do not vary in body size. However, different honeybee species have very different body sizes and this could result in differences in thermal limits. As the authors do not clarify that the honeybee species they refer to is Apis mellifera, this leads to the impression that these results would apply to other honeybee species with different body sizes, but this may not be the case.

Thank you for this thoughtful feedback. We have made changes throughout the text to clarify that we are referring to Apis mellifera, as well as bumblebees specifically from temperate regions. We have also added a paragraph to the discussion about the limitations of using only a few species to parameterise our model. (Lines 383-392)

Line 31, 55 and line 181: the model apparently only accounts for temperature changes and not changes in other weather factors, such as cloud cover, wind or humidity. If this is not the case, then please specify what other weather variables were used, if this is the case then clarify by replacing ‘weather’ here with ’temperature’.

Thank you for pointing this out. Temperature is an important weather variable, but it is not the only weather variable used in the model. The model includes solar radiation, ground surface temperature, air temperature, and relative humidity. We have updated the manuscript to reflect this in lines 44-45 and modified the methods section to include this information and define the weather variables where appropriate. The model variables are all defined in Table 6 of the supporting materials.

Equation 1: define the terms dt and dTth in the text before presenting the model

We have updated the text (lines 56-57) to include this information

Equation 1: please also explain why QH is being added rather than subtracted here. Potential heat transfer from the thorax to the head would reduce the thorax temperature rather than increase it as suggested in this equation. Please clarify the reasoning behind this in the text.

We have added text clarifying this reasoning when the model is introduced (lines 70-73) and in the section on physiological heat flux (lines 131-136).

Line 68: This statement should be supported with an explanation or a citation “and head transfer from the thorax to the head are necessary for bees”. There appears to be only two published studies that explore heat transfer to the head in bees and it appears as though they can regulate this. These works should be cited and the statement revised to reflect the state of our knowledge on this:

Meredith G. Johnson, Jordan R. Glass, Jon F. Harrison; A desert bee thermoregulates with an abdominal convector during flight. J Exp Biol 1 October 2022; 225 (19): jeb244147. doi: https://doi.org/10.1242/jeb.244147

Guadalupe Sepúlveda-Rodríguez, Kevin T. Roberts, Priscila Araújo, Philipp Lehmann, Emily Baird, Bumblebee thermoregulation at increasing temperatures is affected by behavioral state, Journal of Thermal Biology, 2024, 103830,

https://doi.org/10.1016/j.jtherbio.2024.103830.

Thank you for these references! We have updated the text to “...and head transfer from the thorax to the head occur in bees” to be more accurate and included the bumblebee reference (line 66). We added the information about bumblebees to the methods section as well as expanding on our source for heat transfer to the head in honeybees (lines 119-123). The reference on Sonoran desert bees fits well in the second to last paragraph of our discussion, so we have added it there (lin

---

## [Decision Letter · Decision Letter 1]

16 Dec 2024

PONE-D-23-42752R1Thermal limits of bumblebees and honeybees are modulated by different functional traits: predictions of a mechanistic modelPLOS ONE

Dear Dr. MacQueen,

Thank you for submitting your manuscript to PLOS ONE. After careful consideration, we feel that it has merit but does not fully meet PLOS ONE’s publication criteria as it currently stands. Therefore, we invite you to submit a revised version of the manuscript that addresses the points raised during the review process.

There are a few suggestions for additions to the discussion which should be easily incorporated. The paper would be in good shape for acceptance then.

We look forward to receiving your revised manuscript.

Kind regards,

Vivek Nityananda

Academic Editor

PLOS ONE

Journal Requirements:

Reviewers' comments:

Reviewer's Responses to Questions

**Comments to the Author**

1. If the authors have adequately addressed your comments raised in a previous round of review and you feel that this manuscript is now acceptable for publication, you may indicate that here to bypass the “Comments to the Author” section, enter your conflict of interest statement in the “Confidential to Editor” section, and submit your "Accept" recommendation.

Reviewer #1: (No Response)

Reviewer #3: (No Response)

2. Is the manuscript technically sound, and do the data support the conclusions?

Reviewer #1: Yes

Reviewer #3: Yes

3. Has the statistical analysis been performed appropriately and rigorously? 

Reviewer #1: Yes

Reviewer #3: Yes

4. Have the authors made all data underlying the findings in their manuscript fully available?

Reviewer #1: Yes

Reviewer #3: Yes

5. Is the manuscript presented in an intelligible fashion and written in standard English?

Reviewer #1: Yes

Reviewer #3: Yes

6. Review Comments to the Author

Reviewer #1: The authors have thoroughly responded to the previous reviews and the manuscript is now much improved. I am happy to recommend this for publication. However the authors should have a look over to ensure that minor grammatical and other issues are resolved.

A few that I noticed include:

1. L26 – 29: This sentence is too long with four commas bringing together 5 subsections. I would suggest to break down this sentence and other like this into 2 or 3 to make it easier to read.

2. L49: ‘our’ should be capitalised as it is the start of the sentence.

3. L326: Please use the author names to refer to the study at the start of the sentence instead of using the reference number as it makes it harder to read.

4. L339: ‘could be allow’ should be replaced with ‘could allow’

Reviewer #3: This study makes a valuable contribution to understanding the effects of temperature on bees, particularly by presenting a model to measure thorax temperature. Bumblebee thorax temperature is a critical parameter influencing their ability to fly, forage, and survive under varying environmental conditions. The model developed here is an invaluable tool for investigating how bees respond to thermal stress, offering key insights into their adaptability to climate change and other stressors. By defining these physiological limits, we can better predict the resilience of bumblebee populations amidst the increasing frequency and intensity of extreme weather events, such as heatwaves.

Main Comments:

1. Include species names in methods:

While the study includes one honeybee species and several bumblebee species, the names of these species are only mentioned in the Supplementary Material. Including these species in the Methods section would enhance the manuscript's clarity and accessibility. This change would be particularly valuable for researchers focused on the variability in thermal tolerance among bees.

2. Address high CTmax in some bumblebee species:

In L345–348, the discussion suggests that a resting bumblebee would reach lethal thorax temperatures at air temperatures of 42°C. However, some studies indicate that certain bumblebee species (including those from temperate regions) can tolerate air temperatures exceeding 45°C (CTmax) (e.g. https://doi.org/10.1242/jeb.165589;
https://doi.org/10.1016/j.jtherbio.2023.103672). Incorporating these findings into the discussion would provide a more comprehensive view of thermal limits.

3. Consider the role of nutritional status:

It may be worth addressing whether nutritional status could influence an individual bee's flight activity and ability to thermoregulate under different weather conditions, particularly high temperature. This could add a valuable dimension to the study and highlight a potential avenue for future research.

General Feedback:

The authors have done an excellent job addressing reviewers’ concerns with appropriate revisions, thoroughly explaining the model, and discussing the results in depth. I particularly appreciated the discussion on how the model can be applied to other insect species and the thoughtful consideration of its limitations.

Minor Suggestions:

• L20–22: Include the range or minimum thorax temperature required for flight, if available. This would underscore the bees’ remarkable ability to elevate thorax temperature despite ambient conditions.

• L86: Clarify whether "ground temperature" refers specifically to soil temperature or includes other surfaces, such as impervious materials, which are present in urban areas.

• L241: Specify the approximate time required for the thorax to reach equilibrium temperature, as “after some time” is a bit vague.

Additional Minor Comments:

• L55: Capitalise “Our parameters.”

• Figure 1 caption: Correct “icoming” to “incoming.”

• L76: Change “tounge” to “tongue.”

• L245: Correct “activiating” to “activating.”

• Figure 3 caption: Correct “rigth” to “right” and revise the third sentence to: “Pink curves indicate flight, and green curves indicate resting.”

• L319–325: Clarify when referring to body mass or thorax mass.

• L338: Specify whether “higher thermal limit” refers to upper or lower thermal limits.

• L366: Change to “It is dependent.”

• L392: Revise to: “Could allow bees.”

Overall, this manuscript is well-written, thoroughly researched, and presents a model with significant implications for understanding insect physiology under climate change.

7. PLOS authors have the option to publish the peer review history of their article (what does this mean?). If published, this will include your full peer review and any attached files.

Reviewer #1: No

Reviewer #3: No

---

## [Author Response · Author response to Decision Letter 2]

2 Feb 2025

Comments to the Author

Reviewer #1: The authors have thoroughly responded to the previous reviews and the manuscript is now much improved. I am happy to recommend this for publication. However the authors should have a look over to ensure that minor grammatical and other issues are resolved.

A few that I noticed include:

1. L26 – 29: This sentence is too long with four commas bringing together 5 subsections. I would suggest to break down this sentence and other like this into 2 or 3 to make it easier to read.

2. L49: ‘our’ should be capitalised as it is the start of the sentence.

3. L326: Please use the author names to refer to the study at the start of the sentence instead of using the reference number as it makes it harder to read.

4. L339: ‘could be allow’ should be replaced with ‘could allow’

Response: Thank you for pointing these out. We have fixed these errors and carefully read the entire manuscript to fix any other minor grammatical issues.

Reviewer #3: This study makes a valuable contribution to understanding the effects of temperature on bees, particularly by presenting a model to measure thorax temperature. Bumblebee thorax temperature is a critical parameter influencing their ability to fly, forage, and survive under varying environmental conditions. The model developed here is an invaluable tool for investigating how bees respond to thermal stress, offering key insights into their adaptability to climate change and other stressors. By defining these physiological limits, we can better predict the resilience of bumblebee populations amidst the increasing frequency and intensity of extreme weather events, such as heatwaves.

Main Comments:

1. Include species names in methods:

While the study includes one honeybee species and several bumblebee species, the names of these species are only mentioned in the Supplementary Material. Including these species in the Methods section would enhance the manuscript's clarity and accessibility. This change would be particularly valuable for researchers focused on the variability in thermal tolerance among bees.

Response: thank you for this comment. We hadn’t included the species names for bumblebees in the methods because different parameters are drawn from different species (as not all parameters are available for any one species, unfortunately), so the model doesn’t represent any single species but rather a sort of conglomerate bumblebee. We have now provided the list of species as a footnote (from line 50).

2. Address high CTmax in some bumblebee species:

In L345–348, the discussion suggests that a resting bumblebee would reach lethal thorax temperatures at air temperatures of 42°C. However, some studies indicate that certain bumblebee species (including those from temperate regions) can tolerate air temperatures exceeding 45°C (CTmax) (e.g. https://doi.org/10.1242/jeb.165589;
https://doi.org/10.1016/j.jtherbio.2023.103672). Incorporating these findings into the discussion would provide a more comprehensive view of thermal limits.

Response: Thank you for pointing this out. We have incorporated this information into the discussion.

3. Consider the role of nutritional status:

It may be worth addressing whether nutritional status could influence an individual bee's flight activity and ability to thermoregulate under different weather conditions, particularly high temperature. This could add a valuable dimension to the study and highlight a potential avenue for future research.

Response: This is an interesting point, and although well outside the scope of our model it is worth considering. We have added a paragraph in the discussion.

General Feedback:

The authors have done an excellent job addressing reviewers’ concerns with appropriate revisions, thoroughly explaining the model, and discussing the results in depth. I particularly appreciated the discussion on how the model can be applied to other insect species and the thoughtful consideration of its limitations.

Minor Suggestions:

• L20–22: Include the range or minimum thorax temperature required for flight, if available. This would underscore the bees’ remarkable ability to elevate thorax temperature despite ambient conditions.

• L86: Clarify whether "ground temperature" refers specifically to soil temperature or includes other surfaces, such as impervious materials, which are present in urban areas.

• L241: Specify the approximate time required for the thorax to reach equilibrium temperature, as “after some time” is a bit vague.

Additional Minor Comments:

• L55: Capitalise “Our parameters.”

• Figure 1 caption: Correct “icoming” to “incoming.”

• L76: Change “tounge” to “tongue.”

• L245: Correct “activiating” to “activating.”

• Figure 3 caption: Correct “rigth” to “right” and revise the third sentence to: “Pink curves indicate flight, and green curves indicate resting.”

• L319–325: Clarify when referring to body mass or thorax mass.

• L338: Specify whether “higher thermal limit” refers to upper or lower thermal limits.

• L366: Change to “It is dependent.”

• L392: Revise to: “Could allow bees.”

Thank you. We have incorporated the Minor Suggestions and Additional Minor Comments.

---

## [Editor Report · Decision Letter 2]

13 Feb 2025

Thermal limits of bumblebees and honeybees are modulated by different functional traits: predictions of a mechanistic model

PONE-D-23-42752R2

Dear Dr. MacQueen,

We’re pleased to inform you that your manuscript has been judged scientifically suitable for publication and will be formally accepted for publication once it meets all outstanding technical requirements.

Kind regards,

Vivek Nityananda

Academic Editor

PLOS ONE
---

## [Editor Report · Acceptance letter]

PONE-D-23-42752R2

PLOS ONE

Dear Dr. MacQueen,

I'm pleased to inform you that your manuscript has been deemed suitable for publication in PLOS ONE. Congratulations! Your manuscript is now being handed over to our production team.

Kind regards,

on behalf of

Dr. Vivek Nityananda

Academic Editor

PLOS ONE